# Chemical Compositions and Characteristics of Biocalcium from Pre-Cooked Tuna Bone as Influenced by Sodium Chloride Pretreatment and Defatting by Asian Seabass Lipase

**DOI:** 10.3390/foods13081261

**Published:** 2024-04-20

**Authors:** Soottawat Benjakul, Saowakon Pomtong, Afeefah Chedosama, Jirakrit Saetang, Pornsatit Sookchoo, Krisana Nilsuwan

**Affiliations:** 1International Center of Excellence in Seafood Science and Innovation (ICE-SSI), Faculty of Agro-Industry, Prince of Songkla University, Hat Yai, Songkhla 90110, Thailand; soottawat.b@psu.ac.th (S.B.); nampomtong43@gmail.com (S.P.); afeefah.615@gmail.com (A.C.); jirakrit.s@psu.ac.th (J.S.); 2Center of Excellence in Bio-Based Materials and Packaging Innovation, Faculty of Agro-Industry, Prince of Songkla University, Hat Yai, Songkhla 90110, Thailand; pornsatit.s@psu.ac.th

**Keywords:** biocalcium, pre-cooked tuna bone, pretreatment, defatting process

## Abstract

Pre-cooked bone is a waste product generated during tuna processing and can serve as a potential source of biocalcium (BC). Generally, non-collagenous protein and fat must be removed properly from bone. A NaCl solution can be used to remove such proteins, while fish lipase can be used in a green process, instead of solvent, for fat removal. Thus, this study aimed to investigate the impact of NaCl pretreatment at different concentrations in combination with heat to eliminate non-collagenous proteins, and to implement fish lipase treatments at varying levels for fat removal, for BC production from pre-cooked tuna bone. Optimal NaCl pretreatment of bone was achieved when a 5% NaCl solution at 80 °C was used for 150 min. The lowest lipid content was obtained for bone defatted with crude lipase extract (CLE) at 0.30 Unit/g of bone powder for 2 h. BC powder from bone defatted with CLE (DF-BC) possessed greater contents of ash, calcium, and phosphorus and smaller particle sizes than the control BC powder. X-ray diffractograms suggested that both BC powders consisted of hydroxyapatite as a major compound, which had a crystallinity of 62.92–63.07%. An elemental profile confirmed the presence of organic and inorganic matter. Thus, BC powder could be produced from pre-cooked tuna bone using this ‘green process’.

## 1. Introduction

Calcium is crucial for human health. Calcium deficiency is a global issue for humans and is associated with several diseases, i.e., reduction of bone mass, rickets, and osteoporosis [1]. The amount of calcium in many diets is insufficient for the human body’s requirements. To conquer a deficiency of calcium, calcium fortification in foods is a promising approach. Calcium carbonate and calcium citrate are widely used as dietary calcium supplements in the form of tablets [2]. Other available calcium supplements include calcium phosphates, calcium gluconate, calcium fumarate, calcium malate, calcium lactate, and some mixed salts, such as calcium lactate malate or calcium lactate citrate [3]. Nonetheless, its low bioavailability is a drawback [4], leading to calcium deficiency associated with poor absorption [5]. Biocalcium, especially from fish bone, has drawn increasing interest. Collagenous proteins complexed with calcium hydroxyapatite can help upsurge the solubility and bioavailability of calcium [4].

Thailand is one of the world’s largest fish exporters, the scale of which increased from 5.6 to 5.8 billion US dollars from 2016 to 2017 [6]. For the Thai canning industry, skipjack tuna is the most abundant species used for production [7]. Typically, bones from pre-cooked skipjack tuna, accounting for 10–15% of the fish, are generated as byproducts during canned tuna processing [8]. Those bones are rich in calcium (41.2–49.2%) and collagen [9]. Several processing methods, including non-collagenous protein removal, fat removal, bleaching, softening, drying, and grinding, have been developed [4,10]. Alkaline pretreatment with a 2 M NaOH solution (50 °C, 30 min) is generally employed for non-collagenous protein removal [4]. This process is effective for the removal of remaining meat and blood in pre-cooked tuna bone. For fat removal or defatting processes, various solvents such as hexane, isopropanol and their mixture are applied, followed by evaporation to obtain defatted bone with less or no fat [4,10].

However, to avoid the use of chemicals and solvents, alternative approaches could be implemented. Primarily, the remaining meat or non-collagenous protein is mostly salt-soluble protein. Increasing sodium chloride concentration enhances the repulsive force between the protein molecules, leading to augmented protein solubilization [11]. Furthermore, fish bone also contains fats or lipids, which can oxidize and develop an undesirable odor/flavor, especially a fishy odor in biocalcium [4,10]. A defatting process with lipase from fish viscera is a potential alternative for fat removal from fish bone. Additionally, the typically used solvents are excluded, leading to the reduction of capital cost, as the costs of solvents and their residues can be omitted. Furthermore, use of lipase has high selectivity and efficacy, and is environmentally friendly [12]. After ester bonds are hydrolyzed, the free fatty acid and glycerol generated can be washed out, leading to the reduction of fat [12]. Consequently, the remaining lipid is reduced. However, negligible information exists regarding the use of a NaCl solution for non-collagenous protein removal and fish lipase for defatting of pre-cooked tuna bone. Therefore, this investigation aimed to elucidate the effect of NaCl pretreatment on the removal of non-collagenous protein from tuna bone and to investigate the impacts of Asian seabass lipase on fat removal from tuna bone, as well as the yield, composition, and characteristics of the resulting biocalcium powders.

## 2. Materials and Methods

### 2.1. Chemicals

Sodium chloride was acquired from KemAus™ (Phakanong, Bangkok, Thailand). Hydrochloric acid was obtained from QReC (Auckland, New Zealand). The rest of the chemicals used were of analytical grade and were obtained from Sigma Chemical Co. (St. Louis, MO, USA).

### 2.2. Effect of Sodium Chloride in Combination with Heat on Non-Collagenous Protein Removal of Pre-Cooked Tuna Bone

Bones from pre-cooked skipjack tuna (*Katsuwonus pelamis*) were provided from Tropical Canning (Thailand) Public Company Limited (Songkhla, Thailand). The bones (25 g), cut into 5 cm in length, were mixed with distilled water (DW) or 125 mL of NaCl solution at varying concentrations (5 and 10%, *w*/*v*), followed by agitation with an overhead stirrer (RW 20 digital, IKA-Werke GmbH & CO.KG, Staufen, Germany) (150 rpm, 180 min, 80 °C). Solution samples (5 mL) were taken at 0, 15, 30, 45, 60, 90, 120, 150, and 180 min for analyses of total soluble protein content [13] and hydroxyproline content [14]. The condition yielding the highest removal of non-collagenous protein with the highest retained hydroxyproline content in the bone was selected for further investigation.

### 2.3. Impact of Lipase from Asian Seabass Liver on Defatting of Tuna Bone

#### 2.3.1. Preparation and Extraction of Crude Lipase

Crude lipase extract (CLE) from Asian seabass liver was prepared. Firstly, the liver was separated from other internal organs, chopped, and ground in the presence of liquid nitrogen (Model MX-898N, Panasonic, Panasonic Sdn. Bhd., Kuala Lumpur, Malaysia). The sample was defatted using acetone. Acetone-treated powder was used for extraction of lipase and CLE was examined for lipase activity using ρ-nitrophenyl palmitate as a substrate at pH 8 and 50 °C [12]. One unit of activity was defined as the amount of enzyme producing 1 µmol ρ-nitrophenol per min under the assay condition.

#### 2.3.2. Defatting Process of Tuna Bone with CLE

The bones, after the non-collagenous protein removal process using a salt solution, were washed in distilled water (DW) thoroughly. The cleaned bones were dried in a rotary-tray dryer at 60 °C using air velocity of 1.5 m/s for 6 h. The dried bones were reduced in size with a blender (Panasonic, Model MX-898N, Berkshire, UK) for 30 s. The ground bones were dispersed in DW at a bone/DW ratio of 1:5 (*w*/*v*) and the pH was adjusted to 7.0 using a 1 M HCl solution. The CLE was added into sample mixtures at 0.15 and 0.30 Unit/g bone powder, followed by incubation at 30 °C with continuous stirring for various times (0.5, 1, 2, and 3 h). At the designated time, the treated bones were centrifuged (10,000× *g*, 10 min, RT) using a centrifuge (model CR22N, Hitachi, Hitachi Koki Co., Ltd., Tokyo, Japan). The bones were washed with 5 volumes of DW, centrifuged, and dried as described above. A sample was also prepared without a defatting process and named as the control. All sample powders were analyzed.

#### 2.3.3. Lipid Content and Distribution

Lipid content was determined using the Bligh and Dyer method [15]. Lipid distribution in the tuna bone samples was examined using an Olympus confocal microscope (Model FV300, Tokyo, Japan). The bones (50 mg) were dispersed in DW (1:20, *w*/*v*). The mixtures were mixed with 0.01% Nile blue A (500 uL). A 50 μL sample of each mixture was placed on a slide. A fluorescence microscope with an excitation wavelength of 533 nm and an emission wavelength of 630 nm was used. Magnification of 100× was employed.

### 2.4. Production and Characterization of Biocalcium

The bone defatted with lipase at a concentration of 0.30 Unit/g bone for 2 h (DF-Bone) was selected for the production of biocalcium and characterized in comparison with the control without a defatting process (Con-Bone). The dried DF-Bone and Con-Bone were reduced into coarse particles using a blender. Then, a fine particle size was attained with the aid of a ball milling machine (Model PM 100, Retsch GmbH, Haan, Germany). Each sample (200 mL) was poured into a grinding jar with twenty-five grinding balls (ϕ = 20 mm). Grinding was then performed (200 rpm, 2.5 h). Ground bone was then sieved to a particle size less than 75 µm using an electric sieving machine (E.V.S.1., Endecotts Ltd., London, UK). Analyses of all powder samples or BC were conducted.

#### 2.4.1. Yield

The yield was calculated as follows:(1)$$\mathrm{Yield}\;\left(\%\right)=\frac{\mathrm{Dry}\;\mathrm{weight}\;\mathrm{of}\;\mathrm{BC}\;(g)}{\mathrm{Dry}\;\mathrm{weight}\;\mathrm{of}\;\mathrm{initial}\;\mathrm{bone}\;(g)}\times 100$$

#### 2.4.2. Hydroxyproline Content

The spectrophotometric method as described by Bergman and Loxley [14] was employed to examine hydroxyproline content in BC samples. Biocalcium powders were digested with hydrochloric acid at 110 °C for 24 h prior to the addition of activated carbon and then filtered with filter paper. The filtrates were then mixed with isopropanol, oxidation solution, and Ehrlich solution and the mixtures were incubated at 60 °C for 25 min. The mixtures were then diluted and measured for absorbance at 558 nm. The hydroxyproline amount was computed from the hydroxyproline standard curve (0–60 ppm).

#### 2.4.3. Chemical Composition

The moisture, protein, fat, and ash contents of the biocalcium powders were examined using the AOAC method. Analytical method numbers used were No. 925.45, 981.10, 948.15, and 923.03, respectively [16].

#### 2.4.4. Calcium and Phosphorus Contents

The determination of Ca and P in all the samples was carried out with the aid of an inductively coupled plasma optical emission spectrometer (ICP-OES) (Model Optima 4300 DV, Perkin Elmer, Shelton, MA, USA). For the detection of Ca and P, the wavelengths of 317.933 and 213.617 nm were used, respectively [17].

#### 2.4.5. Color

A Hunterlab Colorfex EZ colorimeter (Hunter Associates Laboratory Inc., Reston, VA, USA) was employed for the measurement of color values of BC samples. Lightness (*L**), redness/greenness (*a**), and yellowness/blueness (*b**) were recorded. The differences in color (Δ*E**) were also computed [10].

#### 2.4.6. Mean Particle Size (MPS)

MPS was determined using a laser particle size analyzer (LPSA) (Model LS 230, Beckman Coulter^®^, Fullerton, CA, USA) [4]. After the dispersed sample in DW was placed in the device, 5 consecutive measurements were conducted. Volume-weighted mean particle diameter (*d*_43_) was then computed.

#### 2.4.7. X-ray Diffraction (XRD)

The phase composition of the powder samples was assessed by XRD [9]. An X-ray Diffractometer (WI-RES-XRD EMPREYAN-001, Panalytical, The Netherlands) was used at the wavelength of 0.154 nm (Cu K-α radiation) with the specified conditions [9]. Phase identification was carried out and data were interpreted [9].

#### 2.4.8. Scanning Electron Microscopy with Energy Dispersive X-ray Spectroscopy (SEM–EDX)

A scanning electron microscope (SEM) equipped with an electron-dispersive X-ray spectroscope (EDX) (SEM-EDX) (X-Max80; Oxford, UK) was used [10]. The samples were coated with gold and determined with the secondary electron mode at a 20 kV accelerating voltage. Elemental composition was examined via the elemental mapping analysis mode. Elemental mapping images were captured and recorded at a magnification of 1000× using AZtec Software (version 2.4, Oxford Instruments plc, Abingdon, UK).

### 2.5. Statistical Analysis

Completely randomized design (CRD) was used. Experiments and analyses were done in triplicate. Analysis of variance (ANOVA) was carried out and Duncan’s multiple range test at the *p* < 0.05 level was employed for difference analysis. Pair sample *t*-tests were conducted for pair comparison. The SPSS package (SPSS for windows, Version 28, SPSS Inc., Chicago, IL, USA) was utilized.

## 3. Results and Discussion

### 3.1. Impact of Sodium Chloride in Combination with Heat on Non-Collagenous Protein Removal of Pre-Cooked Tuna Bone

Proteins from pre-cooked tuna bones were solubilized in NaCl solutions at varying concentrations (0, 5, and 10%) for different times (Figure 1A). Generally, the heating process enhanced the release of protein, especially as heating time increased, as witnessed by the increased total soluble protein (TSP) content in solution (*p* < 0.05). When the protein was heated, degradation plausibly took place [18]. The liberation of small proteins or peptides from the bone matrix could occur, leading to the rise of TSP content in the water. Protein, hydroxyproline, and α-amino group contents in the water rose when tuna bone was exposed to high-pressure heating at 121 °C, especially with an increase in heating time from 30 to 90 min [9]. Moreover, TSP content in the NaCl solution increased with longer treatment times (*p* < 0.05), irrespective of NaCl concentration. Greater TSP content (*p* < 0.05) was noted in both NaCl solutions used for treatment of bones than that found in DW when the same time was employed (*p* < 0.05). Generally, TSP drastically rose in the first 15 min, especially when both NaCl solutions were used. In general, a continuous upsurge in TSP was found for both NaCl solutions after 30 min up to 180 min. No differences in TSP between the solutions containing NaCl at both concentrations at all treatment times were observed (*p* > 0.05). NaCl was able to solubilize the proteins localized in bones or attached to the bones, resulting in the elimination of proteinaceous constituents from the bones. Proteins were solubilized at a higher extent with increased treatment time, most likely associated with the enhanced mass transfer of partially denatured or solubilized proteins from the bones.

Figure 1B illustrates the hydroxyproline content, which is a unique amino acid present in collagen, throughout the NaCl pretreatment. Typically, bones are composed of hydroxyapatite, which is localized between gap zones of collagen fibrils [19]. Upon heating, the protein is solubilized, known as thermal solubilization [20]. It is worth noting that the liberation of peptides or proteins, particularly collagen, from the bone matrix led to the rise of the amount of HYP in the pretreatment solutions. Generally, collagenous proteins were continuously released until 180 min for all pretreatment processes. After 30 min of pretreatment, HYP content (*p* < 0.05) was increased in the NaCl solutions at both levels. Since 5% NaCl for 150 min could remove proteins from and maintain collagen in bone effectively, it was implemented for the pretreatment of pre-cooked tuna bone.

### 3.2. Effect of Lipase from Asian Seabass Liver on Defatting of Tuna Bone

#### 3.2.1. Lipid Content

Lipid content in NaCl-pretreated tuna bones subjected to hydrolysis with crude lipase extract (CLE) at varying levels and incubation times is depicted in Figure 2. The highest lipid content was observed for the control without a defatting process (*p* < 0.05). For the same incubation time, a lower lipid content was attained in bone treated with lipase at a higher level (*p* < 0.05). The effectiveness of lipase in eliminating lipid from tuna bone typically rose as longer incubation times were used (*p* < 0.05). A lower content of lipid was observed in tuna bone hydrolyzed by lipase at 0.30 Unit/g bone for 2 and 3 h (*p* < 0.05), in which lipid reductions of 36.04–36.94% were found. Glycerol and free fatty acids might be eliminated during further washing with water. Lipase can cleave ester bonds in triacylglycerols or phospholipids [12]. Lipids containing a high amount of unsaturated fatty acids (PUFAs) undergo auto-oxidation with ease, causing deterioration. As a result, toxic compounds, undesirable color and odor/flavor, and nutritional losses are noticeable. This also renders food less appealing to consumers. Lipase from Asian seabass liver was shown to have high efficiency in defatting tuna bone, and therefore the use of solvent could be omitted.

#### 3.2.2. Lipid Distribution

Figure 3 depicts the distribution of lipids in tuna bone, both without and with defatting using lipase at various levels and times, as observed using a confocal laser scanning microscope (CLSM). All tuna bones defatted with CLE had fewer residual lipids in comparison with that of the control (non-defatted bone). This result suggested that the lowest amount of lipids was retained for tuna bone treated with CLE (0.30 Unit/g bone powder) with a longer incubation time (more than 2 h). Less lipid distribution from bone treated with CLE aligns with the lowered lipid content documented in Figure 2. Fish bone typically contains lipids in its matrix [9]. Lipase in CLE might hydrolyze lipids, and the resulting free fatty acids and glycerol were eliminated by washing. This was witnessed by the reduced lipid distributed in CLE-treated bone, as illustrated in CLSM micrographs (Figure 3).

### 3.3. Yield, Composition, and Characteristics of Biocalcium

#### 3.3.1. Yield

The yields of BC powders from the bones without (Con-BC) and with a defatting process (DF-BC) are shown in Table 1. The Con-BC sample showed a higher yield (86.56%) than that of DF-BC (73.01%) (*p* < 0.05). This was likely associated with higher loss of lipid and some loss of proteins, peptides, and smaller size of hydroxyapatite during the defatting process and washing. Thus, the defatting process exhibited an adverse effect on the yield of biocalcium.

#### 3.3.2. Hydroxyproline Content

Table 1 displays the hydroxyproline (HYP) contents of Con-BC and DF-BC powders. The HYP contents of Con-BC and DF-BC powders were 12.86 and 15.21 mg/g of dry sample, respectively. DF-BC had a higher hydroxyproline content (*p* < 0.05) than that of Con-BC. The defatting process could facilitate the removal of lipid from the bone, as ascertained by the augmented hydroxyproline content in the resulting BC powder. Coincidentally, a lower lipid content in bone after the defatting process was demonstrated in Figure 2. HYP content was therefore upsurged by the defatting process using CLE.

#### 3.3.3. Chemical Compositions

The chemical compositions of both BC powders are shown in Table 1. Moisture, protein, fat, and ash contents in both BC powders are in the range of 3.08–5.33, 24.57–27.77, 3.76–5.46, and 58.66–61.43%, respectively. Protein contents of Con-BC and DF-BC were in line that found in biocalcium from pre-cooked skipjack tuna bones (24.26%) [4]. Nonetheless, both BC powders had higher fat content along with lower ash content than those defatted by solvent (0.21% and 72.20%, respectively) [4]. Moreover, DF-BC powder showed lower moisture (3.08%), protein (24.57%), and fat (3.76%) contents than those of Con-BC powder (5.33%, 27.77%, and 5.46%, respectively) (*p* < 0.05). The defatting process with CLE could thus reduce fat or lipid in the bone powder, resulting in a higher concentration of ash content in the bone matrix. However, some retained proteases in CLE could hydrolyze the protein in the bone, resulting in the lower protein content of the DF-BC sample. The higher ash content (61.43%) with lower fat content (3.76%) were noticeable for the BC powder from the bone powder defatted with CLE (*p* < 0.05). Therefore, the defatting process had an influence on chemical compositions of obtained BC powder.

#### 3.3.4. Calcium and Phosphorus Contents

The calcium and phosphorus contents of BC powders prepared from the bones without (Con-BC) and with a defatting process (DF-BC) are shown in Table 1. The Ca and P contents were different between both samples (*p* < 0.05). The DF-BC powder possessed greater calcium (26.18%) and phosphorus (14.51%) contents (*p* < 0.05) than those of Con-BC powder (23.66% and 14.07%, respectively). This was in line with the higher ash content in the former, as denoted in Table 1. Since the tuna bone after the defatting process still had a lower amount of remaining organic substances, particularly lipid, a higher proportion of minerals was achieved. The defatting process resulted in the liberation of lipids from tuna bone, thereby elevating the proportion of inorganic matter.

The Con-BC and DF-BC powders exhibited mole ratios of Ca and P at 1.30 and 1.39, respectively. Hydroxyapatite (HAP) was typically documented as the main mineral phase in fish bones [21,22]. Pure hydroxyapatite consists of 39.8% Ca and 18.5% P, with a Ca/P mole ratio of 1.67 [23]. The Con-BC and DF-BC powders had Ca/P mole ratios lower than 1.67. This might be due to the presence of Ca in other forms, apart from Ca-hydroxyapatite. Additionally, a higher mole ratio was obtained for the DF-BC powder, compared to that of the Con-BC powder. The purity of fish bones might govern the Ca/P ratios of biological apatites [21,24]. The defatting process might enlarge the co-elution of P along with organic matter such as lipids, leading to a higher relative Ca/P mole ratio of HAP in the obtained BC powder. The Ca/P mole ratio of the BC powders from the tuna bone exhibited a resemblance to the ratio of BC powder manufactured from other fish bones (1.29–1.62) [25]. Overall, different amounts of calcium, phosphorus, and other organic matter can be detected between different fish bones.

#### 3.3.5. Color

The BC powder from tuna bone defatted with CLE (DF-BC) had a lower *L**-value (*p* < 0.05) but similar *a**-, *b**-, and Δ*E**-values to the BC powder from the bone without a defatting process (Con-BC) were noted (Table 1). It is worth noting that the Con-BC powder with a whitish creamy color might be influenced by light scattering. The higher amount of remaining fat droplets could bring about light scattering to a higher extent, resulting in the high lightness value in the Con-BC sample. Nevertheless, the defatting process had no effects on the redness, yellowness, and total difference in color (Figure 4). Therefore, the defatting process slightly lowered the lightness of the BC powder prepared from bone defatted by lipase from Asian seabass liver.

#### 3.3.6. Mean Particle Size (MPS)

Both BC powders exhibited distinct MPSs. The DF-BC powder displayed a smaller MPS than the Con-BC powder (*p* < 0.05) (Table 1). The defatting process could loosen the bone structure to some degree. As a result, grinding could be performed with ease and a smaller MPS was attained. The agglomeration and stickiness of the particles in the powders might be governed by the existence and integrity of the bone matrix. The size of particles is an important factor affecting the properties and sensation of food products, including their texture, appearance, and aroma [26]. The mouthfeel and overall acceptability of products fortified with biocalcium from skipjack tuna eyeball scleral cartilage are mainly impacted by the size of particles in biocalcium powder.

#### 3.3.7. X-ray Diffraction (XRD) Diffractograms

Figure 5 shows the X-ray diffractograms of the Con-BC and DF-BC powders. Both BC powders exhibited diffraction peaks corresponding to the crystalline phase of HAP (ICDD: 01-074-4172) at angles 25.95°, 31.69°, 33.07°, 39.85°, 46.66°, and 64.15°. Moreover, both BC powders displayed broad peaks, owning to the inelastic and elastic scattering of HAP nanocrystals [27]. Less intensity and broader peaks were obtained for both BC powders, attributed to the presence of extracellular matrix and proteins involving collagen [9]. Furthermore, the DF-BC powder exhibited a higher intensity and narrower full width at half maximum (FWHM) of diffraction peaks, indicating the clustering of nano-hydroxyapatite to the larger crystals. The defatting could therefore eliminate some organic matter, especially fat, from the bone matrix, thereby favoring the formation of HAP crystal agglomeration, as ascertained by the higher crystallinity. The crystallinity of both BC powders ranged from 62.92 to 63.07%. Higher crystallinity was found for the DF-BC powder. It is worth noting that a well-crystallized HAP phase was associated with greater crystallinity and phase purity of BC powder when a defatting process was applied.

#### 3.3.8. Scanning Electron Microscopy with Energy Dispersive X-ray Spectroscopy (SEM–EDX)

Scanning electron microscopic images of Con-BC and DF-BC samples are shown in Figure 6. Both biocalcium powders showed a similar microstructure. An irregular shape was observed for both BC powders. However, the DF-BC had the smaller and finer particles, compared to that of Con-BC. This suggested that the defatting process altered the bone structure, which could facilitate the size reduction of particles in BC and allow them to became more uniform and finer.

The elemental contents of Con-BC and DF-BC powders, as determined by SEM-EDX, are illustrated in Figure 6. Both BC powders had oxygen and carbon contents as their major components of organic matter. These types of organic matter might be associated with the presence of protein, especially collagen and remaining fat. Ca and P were observed as the major components of inorganic matter. Magnesium, sodium, and sulfur were also found as minor components of inorganic matter. The types of inorganic matter were related with the minerals, especially Ca and P, which constituted the major components in the hydroxyapatite of biocalcium. Ca and P in both samples were in the range of 15.4–16.5% and 8.1–8.6%, respectively. The obtained Ca and P contents were different from the result indicated in Table 1. This might be related to the sensitivity of SEM-EDX, which varies from 1 to 10%wt, thus causing difficulty in detecting low-Z elements and spectral resolution [28]. Despite the instrumental limitations, the content of C was decreased in the DF-BC sample, reconfirming the elimination of organic compounds like lipids as a result of defatting process.

## 4. Conclusions

Pretreatment with a 5% NaCl solution at 80 °C for 150 min was suitable for eliminating non-collagenous protein from pre-cooked tuna bones. The defatting process with crude lipase extract (CLE) from Asian seabass liver at 0.30 Unit/g bone for 2 h showed the highest efficiency at reducing fat in bone powder. The defatting process had an influence on the yields and characteristics of the BC powders. The defatted BC powder showed higher calcium and phosphorus contents, as well as finer particle sizes, than its non-defatted counterpart. Therefore, defatting using lipase from Asian seabass liver could be used as a green method of BC production.

## Figures and Tables

**Figure 1 foods-13-01261-f001:**
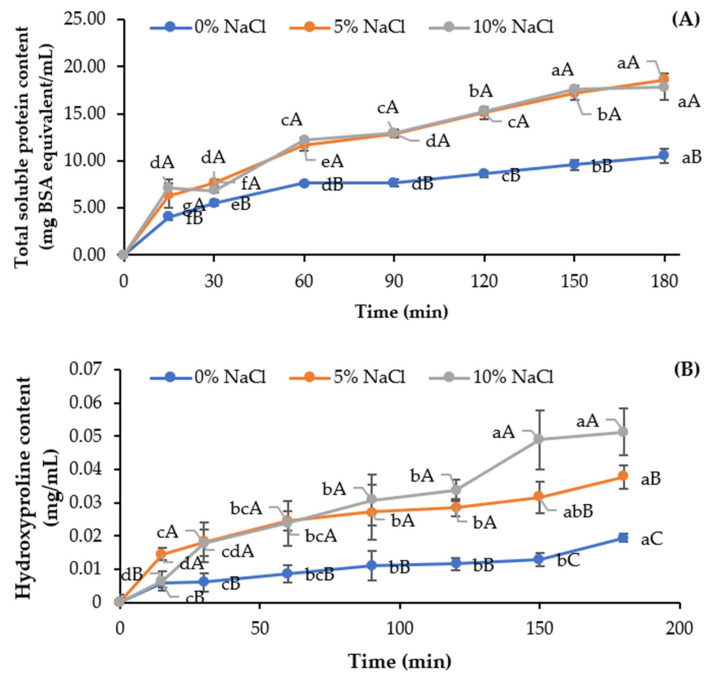
Total soluble protein (**A**) and hydroxyproline (**B**) contents solubilized from pre-cooked tuna bones in NaCl solution at different concentrations as a function of pretreatment time. Bars represent the standard deviations (*n* = 3). Same lowercase letter within the same NaCl concentration denotes non-significant difference (*p* > 0.05). Same uppercase letter within the same pretreatment time denotes non-significant difference (*p* > 0.05).

**Figure 2 foods-13-01261-f002:**
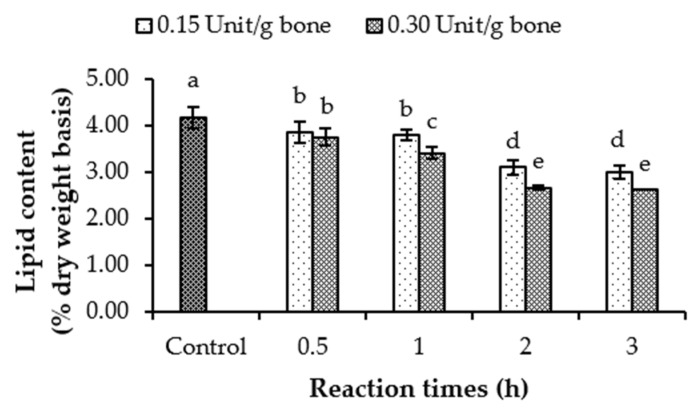
Lipid content of tuna bones defatted with lipase from Asian seabass liver at different levels and reaction times. Bars represent the standard deviations (*n* = 3). Same lowercase letter within the same NaCl concentration denotes non-significant difference (*p* > 0.05).

**Figure 3 foods-13-01261-f003:**
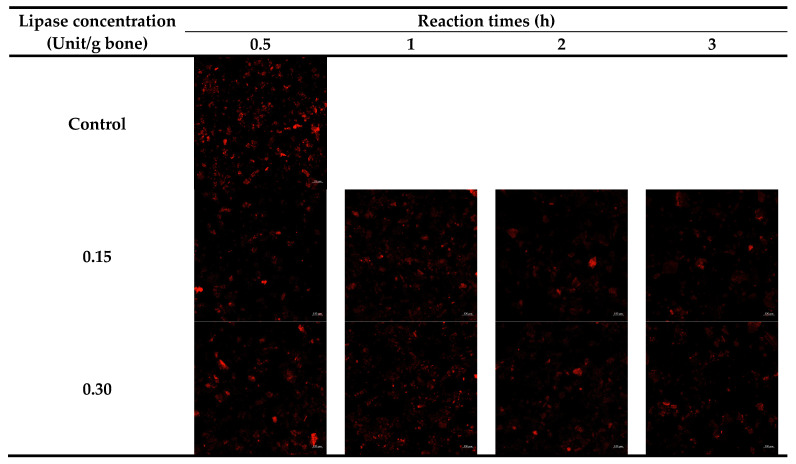
Confocal laser scanning microscopic images of tuna bones defatted with lipase from Asian seabass liver at different levels and reaction times.

**Figure 4 foods-13-01261-f004:**
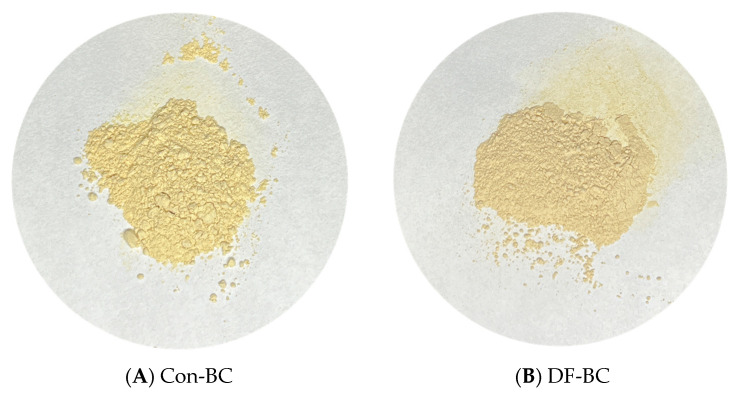
Photographs of biocalcium powders from tuna bone powder without (**A**) and with (**B**) a defatting process. Con-BC and DF-BC: BC powders from tuna bone powders without and with a defatting process, respectively.

**Figure 5 foods-13-01261-f005:**
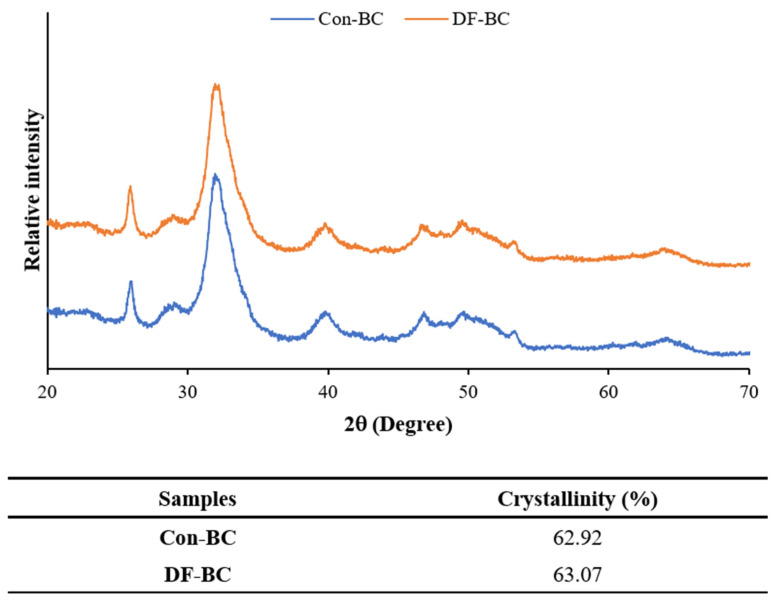
X-ray diffraction (XRD) spectra and crystallinities of biocalcium powders from tuna bone powder without and with a defatting process. Con-BC and DF-BC: BC powders from tuna bone powders without and with a defatting process, respectively.

**Figure 6 foods-13-01261-f006:**
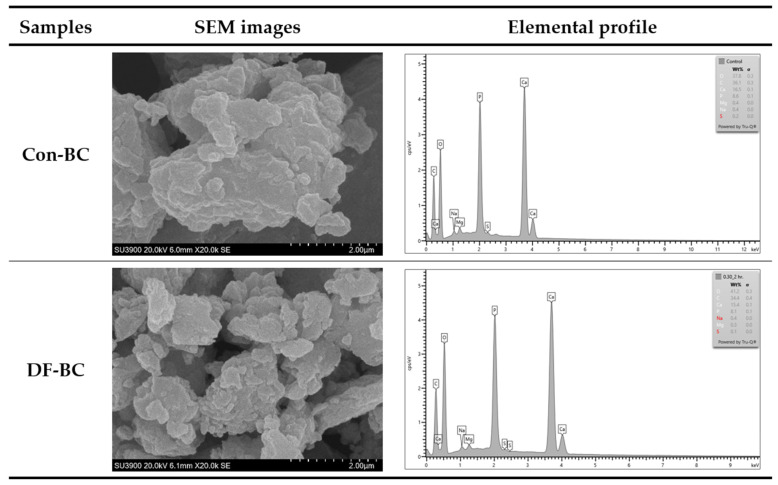
SEM images (magnification at 20,000×) and elemental profile of biocalcium powders from tuna bone powder without and with a defatting process. Con-BC and DF-BC: BC powders from tuna bone powders without and with a defatting process, respectively.

**Table 1 foods-13-01261-t001:** Yield, chemical compositions, colors, and mean particle size diameters of biocalcium powders from tuna bone powder without and with defatting process.

Parameters	Samples	
	Con-BC	DF-BC
Yield (%, dry weight basis)	86.56 ± 2.54 * a	73.01 ± 2.95 b
Hydroxyproline (mg/g dry sample)	12.86 ± 1.00 b	15.21 ± 0.33 a
Moisture content (%, wet weight basis)	5.33 ± 0.04 a	3.08 ± 0.01 b
Protein content (%, dry weight basis)	27.77 ± 0.16 a	24.57 ± 0.05 b
Fat content (%, dry weight basis)	5.46 ± 0.03 a	3.76 ± 0.17 b
Ash content (%, dry weight basis)	58.66 ± 0.33 a	61.43 ± 0.01 b
Calcium (Ca) content (%, dry weight basis)	23.66 ± 0.60 b	26.18 ± 0.17 a
Phosphorus (P) content (%, dry weight basis)	14.07 ± 0.08 b	14.51 ± 0.19 a
Ca/P mole ratio	1.30	1.39
*L**	84.44 ± 0.78 a	83.21 ± 0.70 b
*a**	1.51 ± 0.34 a	1.90 ± 0.27 a
*b**	15.24 ± 1.29 a	15.54 ± 1.38 a
Δ*E**	17.19 ± 1.54 a	18.13 ± 1.56 a
Mean particle size (μm)	15.33 ± 0.40 a	14.03 ± 0.23 b

* Values are presented as mean ± SD (*n* = 3). Same lowercase letter in the same row denotes non-significant differences (*p* > 0.05). Con-BC and DF-BC: BC powders from tuna bone powders without and with defatting process, respectively.

## Data Availability

The original contributions presented in the study are included in the article, further inquiries can be directed to the corresponding author.

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
