# Peer review of "Chemical Compositions and Characteristics of Biocalcium from Pre-Cooked Tuna Bone as Influenced by Sodium Chloride Pretreatment and Defatting by Asian Seabass Lipase"

_foods, 2024, doi:10.3390/foods13081261_

Round 1

Reviewer 1 Report

Comments and Suggestions for Authors

The paper studies the impact of NaCl pretreatment at different concentrations in combination with heat to eliminate non-collagenous proteins and to implement fish lipase at varying levels on fat removal. The results provide method reference and basic data for comprehensive utilization of Tuna bone materials.

I have some points for suggestion:

Line 24 62.92-63.07% provides the number of significant digits, and error.

Line 25 Please provide specific important organic and inorganic matters.

Line 39-66. Can be divided into two paragraphs, one discussing the progress of processing technology, and the other introducing the work of this paper.

Please provide a brief description of these methods such as Hydroxyproline content.

From figure 1, the difference between 5% and 10% NaCl treatment is not obvious. Why not use a more dilute concentration?

Is there any deviation for point "3" in Figure 2?

Please check the valid numbers for the data in the table. For example, the two decimal places of yield are of little significance because the error is in single digits.

Author Response

Responses to reviewer

The paper studies the impact of NaCl pretreatment at different concentrations in combination with heat to eliminate non-collagenous proteins and to implement fish lipase at varying levels on fat removal. The results provide method reference and basic data for comprehensive utilization of Tuna bone materials. I have some points for suggestion:

***Thank you for insightful comments and suggestions. All queries have been responded and the corrections have been done as highlighted in yellow color.

  1. Line 24 62.92-63.07% provides the number of significant digits, and error.

***The same significant numbers were used. However, there was no error of crystallinity value. Those values were obtained from the single scan of x-ray diffraction of each sample. Sorry for this.

  1. Line 25 Please provide specific important organic and inorganic matters.

***Thank you very much for valuable suggestion. Organic matters, including oxygen and carbon, might be mainly associated with the presence of protein, especially collagen and remaining fat in biocalcium powder. Moreover, the inorganic matters such as calcium and phosphorus were as the major constituents in hydroxyapatite in biocalcium. Those details have been provided in the text, which cannot be added into abstract due to the limitation of word count (200 words). Please see line 361 – 362 and 363 – 365.

  1. Line 39-66. Can be divided into two paragraphs, one discussing the progress of processing technology, and the other introducing the work of this paper.

***Thank you very much for your suggestion. The introduction has been modified as per the reviewer’s comment. One paragraph was discussed with the progress of processing technology such as situation of tuna in Thailand and typical processes for production of tuna biocalcium. Another paragraph covers other approaches for production of biocalcium using sodium chloride for solubilization of non-collagenous protein and the use of lipase from fish liver for defatting process. Please see line 42 – 53 and 54 – 70.

  1. Please provide a brief description of these methods such as Hydroxyproline content.

***Actually, the details in materials and methods were shortened to avoid plagiarism. However, a brief method of hydroxyproline content has been added following the reviewer’s suggestion. Please see line 131 – 136.

  1. From figure 1, the difference between 5% and 10% NaCl treatment is not obvious. Why not use a more dilute concentration?

***Thank you for valuable comments. Based on the data in Figure 1, no difference in total soluble protein content was observed between 5% and 10% sodium chloride (NaCl) solution at all reaction times (Figure 1A). However, lower hydroxyproline content was found in 5% NaCl after 150 min, compared to that of 10% NaCl (Figure 1B). This result suggested that 5% NaCl solution can remove non-collagenous protein from tuna bone more effectively than 10% NaCl solution. On the other hand, the lower loss of hydroxyproline was observed in 10% NaCl solution. Therefore, 5% NaCl for 150 min could remove proteins but maintain collagen in bone effectively.

From our preliminary study, the use of NaCl solution at lower concentration, which provided lower ionic strength, was not efficient for removal of non-collagenous proteins from tuna bone properly.

  1. Is there any deviation for point "3" in Figure 2?

***Thank you for valuable suggestions. The standard deviations (SD) for point ‘3’ have been re-checked and provided. The SD of 0.30 Unit/g bone for 3 h was 0.05. Therefore, it is hard to appear clearly as error bar. Please see Figure 2 line 232.

  1. Please check the valid numbers for the data in the table. For example, the two decimal places of yield are of little significance because the error is in single digits.

***Thank you very much for insightful comments. It is true that two digits are of little significance. However, table 1 consisted of many parameters, in which the two digits were used for all data for the consistency of decimal used. Thus, we prefer to use two digits for all values in Table 1.

Reviewer 2 Report

Comments and Suggestions for Authors

The authors present a study that focused on the possible use of waste bone products as a calcium supplement. The idea is interesting, however, a lot of data should be more detailed present. In my opinion, major revision is needed. Detailed comments are listed below

Abstract

- please add general information about all used analytical techniques

- what do you mean by "which had crystallinity of 62.92 – 63.07%."?

Introduction

- please add some data about calcium supplements

- references should be generally updated due to the fact that some of them are quite old. For introduction add min 5 - 10 current references.  Furthermore please add some details about calcium supplements available on the market.

Materials and methods

- did you use purified NaCl or just as delivered?

- The description of the 2.4.2 Hydroxyproline content is not meaningful Please add a detailed description same for 2.4.3 Chemical composition

- for each technique give details of the used apparatus

- what did you investigate using DLS? What kind of particles?

Results and discussion

- samples of bone for CLSM should be presented

- please add some images that will show how the samples look first and after processing

- how the samples were prepared for the studies presented in Table 1? How did you prepare the powders?

- in the results, hydroxyproline was mainly described. What about hydroxyapatite which was highlighted in the abstract? Which compound is more desirable and why?

- I could not find the results which will show/prove that the optimization process was done.

Comments on the Quality of English Language

Intensive editing is recommended

Author Response

Responses to reviewer

The authors present a study that focused on the possible use of waste bone products as a calcium supplement. The idea is interesting, however, a lot of data should be more detailed present. In my opinion, major revision is needed.

***Your understanding in our work is highly appreciated. All queries raised by reviewer have been responded and the corrections in the text have been made as highlighted in green color.

Detailed comments are listed below

  1. Abstract

- please add general information about all used analytical techniques

***Sorry. Analytical techniques cannot be provided in the abstract as per the reviewer’s suggestions. This was due to the limitation of word counts (200 words).

- what do you mean by "which had crystallinity of 62.92 – 63.07%."?

***Type of compound and the structure of biocalcium (BC) powder were analyzed by X-ray Diffractometer. The result showed that BC powder mainly consisted of hydroxyapatite, in which its structure was arranged like crystalline or ordered phase with 62.92 – 63.07% and the rest was of amorphous phase. This value was obtained from software, which divided the area of crystalline peak with total area of all peaks and multiplied with 100 to report as percentage. The higher value, the more crystalline structure was present in the tested samples.

  1. Introduction

- please add some data about calcium supplements

***Some details on calcium supplements have been provided in the text following the reviewer’s suggestion. Please see line 33 – 37.

- references should be generally updated due to the fact that some of them are quite old. For introduction add min 5 - 10 current references.  Furthermore, please add some details about calcium supplements available on the market.

***Actually, the recently published information had been provided in the introduction (2021 – 2023: 5 papers), which were also related to the scope of present study. Moreover, some details of calcium supplements available in the market have been provided in the text as mentioned above.

  1. Materials and methods

- did you use purified NaCl or just as delivered?

***For practical point of view, we used NaCl powder available in the market (Table salt) for our experiment.

- The description of the 2.4.2 Hydroxyproline content is not meaningful Please add a detailed description same for 2.4.3 Chemical composition

***The details in materials and methods were shortened to avoid plagiarism. However, a brief method of hydroxyproline content has been added following the reviewer’s suggestion. Please see line 131 – 136.

Furthermore, the description of 2.4.3 Chemical composition was analyzed using the standard AOAC methods. These methods are commonly known. For the ease to follow, the analytical method number had been already provided for each analyte. Therefore, we prefer to describe it in the current form. Sorry for the inconvenience.

- for each technique give details of the used apparatus

***Thank you for suggestions. Some details have been added to the text. Please see line 104 – 105 and 143 – 144.

- what did you investigate using DLS? What kind of particles?

***Mean particle size in biocalcium powder was determined using laser particle size analyzer (LPSA), which is not the dynamic light scattering (DLS). The result of mean particle size was calculated and presented as volume-weighted diameter (d43). Please see the details in line 152 – 155.

  1. Results and discussion

- samples of bone for CLSM should be presented

***Actually, the confocal laser scanning microscopic (CLSM) images of bone without defatting process had been provided as the control (Figure 3), which represent the lipid distribution in general bone without defatting process.

- please add some images that will show how the samples look first and after processing

***Thank you very much for your suggestions. The photographs of biocalcium powders from tuna bone powder without and with defatting process have been provided in the text as reviewer’s suggestions. Please see line 319.

- how the samples were prepared for the studies presented in Table 1? How did you prepare the powders?

***Bone powders without and with defatting process were dried and ground with a blender to obtain coarse particle. Thereafter, the coarse powders were further ground with a ball milling machine using a grinding jar containing 25 grinding balls (diameter = 20 mm) at rotation speed of 200 rpm for 2.5 h. Ground bone was then sieved to the size less than 75 µm using an electric sieving machine. This information had been provided in the text. Please see section 2.4 line 117 – 125.

- in the results, hydroxyproline was mainly described. What about hydroxyapatite which was highlighted in the abstract? Which compound is more desirable and why?

***The description of hydroxyapatite (HAP), especially in term of crystallinity had been mentioned in section 3.3.7. Please see line 332 – 346. Basically, biocalcium consisted both collagen and HAP in term of complex. Collagen attached to HAP has been known to increase the solubility as well as bioavailability in gastrointestinal tract. As a result, the pretreatment must minimize the loss of collagen as measured by hydroxyproline, the unique amino acid in collagen.

- I could not find the results which will show/prove that the optimization process was done.

***The effect of sodium chloride (NaCl) in combination with heat on non-collagenous protein removal of pre-cooked tuna bone had been documented in Figure 1. The use of 5% NaCl solution under heating at 80°C for 150 min was optimal condition for non-collagenous protein removal, as indicated by high total soluble protein (TSP) content in soaking solution. Moreover, the effect of lipase from Asian seabass liver at different levels (0 – 0.30 Unit/g bone powder) on defatting of tuna bone was also optimized. The defatting process with lipase at 0.30 Unit/g bone powder for 2 h showed the lower fat content (Figure 2 – 3), which was optimal condition for defatting process of tuna bone powder. Additionally, the chemical composition and characteristics of biocalcium powder prepared from selected defatted bone in comparison with biocalcium without defatting process had been done (Table 1, Figure 4 – 6). All details showed the optimization process for the production of biocalcium from tuna bone via green process.
